# The Role of Alpha-Synuclein and Other Parkinson’s Genes in Neurodevelopmental and Neurodegenerative Disorders

**DOI:** 10.3390/ijms21165724

**Published:** 2020-08-10

**Authors:** C. Alejandra Morato Torres, Zinah Wassouf, Faria Zafar, Danuta Sastre, Tiago Fleming Outeiro, Birgitt Schüle

**Affiliations:** 1Department Pathology, Stanford University School of Medicine, Stanford, CA 94304, USA; aletor1@stanford.edu (C.A.M.T.); fzafar@stanford.edu (F.Z.); danutasastre@aol.com (D.S.); 2German Center for Neurodegenerative Diseases, 37075 Göttingen, Germany; Zinah.Wassouf@dzne.de (Z.W.); touteir@gwdg.de (T.F.O.); 3Department of Experimental Neurodegeneration, Center for Biostructural Imaging of Neurodegeneration, University Medical Center Göttingen, 37075 Göttingen, Germany; 4Max Planck Institute for Experimental Medicine, 37075 Göttingen, Germany; 5Translational and Clinical Research Institute, Faculty of Medical Sciences, Newcastle University, Framlington Place, Newcastle Upon Tyne NE2 4HH, UK

**Keywords:** alpha-synuclein, *SNCA*, *PARK2*, 22q11.2 deletion syndrome, autism spectrum disorders, Parkinson’s disease, neuronal development, neurodegeneration, synaptic dysfunction

## Abstract

Neurodevelopmental and late-onset neurodegenerative disorders present as separate entities that are clinically and neuropathologically quite distinct. However, recent evidence has highlighted surprising commonalities and converging features at the clinical, genomic, and molecular level between these two disease spectra. This is particularly striking in the context of autism spectrum disorder (ASD) and Parkinson’s disease (PD). Genetic causes and risk factors play a central role in disease pathophysiology and enable the identification of overlapping mechanisms and pathways. Here, we focus on clinico-genetic studies of causal variants and overlapping clinical and cellular features of ASD and PD. Several genes and genomic regions were selected for our review, including *SNCA* (alpha-synuclein), *PARK2* (parkin RBR E3 ubiquitin protein ligase), chromosome 22q11 deletion/DiGeorge region, and *FMR1* (fragile X mental retardation 1) repeat expansion, which influence the development of both ASD and PD, with converging features related to synaptic function and neurogenesis. Both PD and ASD display alterations and impairments at the synaptic level, representing early and key disease phenotypes, which support the hypothesis of converging mechanisms between the two types of diseases. Therefore, understanding the underlying molecular mechanisms might inform on common targets and therapeutic approaches. We propose to re-conceptualize how we understand these disorders and provide a new angle into disease targets and mechanisms linking neurodevelopmental disorders and neurodegeneration.

## 1. Introduction of Clinical Aspects of Parkinson’s Disease (PD) and Autism Spectrum Disorders (ASDs)

Early onset neurodevelopmental and late-onset neurodegenerative disorders present as entities that appear quite distinct. However, recent evidence has uncovered surprising commonalities and converging features at the clinical, genomic, and molecular level between these two disease spectra and, in particular, between ASD and PD. 

PD is a progressive neurodegenerative disorder characterized by cardinal motor features, including resting tremor, rigidity, postural instability, and bradykinesia. In addition, PD may also present with a wide range of non-motor symptoms, such as autonomic dysfunction, cognitive impairment, sleep disturbances, and neuropsychiatric symptoms, including depression, anxiety, and repetitive or obsessive–compulsive behaviors [1,2,3]. The prevalence of PD is about 1% over the age of 65. More men than women are diagnosed with sporadic PD and this ratio increases with age; while the incidence ratio is comparable in males and females under 50 years (male:female ratio < 1.2), it increases to 1.6 above 80 years [4]. The neuropathological changes that underlie the Lewy body disease (LBD) spectrum include PD, dementia with Lewy bodies (DLB), and Parkinson’s disease dementia (PDD) [5]. These diseases are characterized by the accumulation of intracellular protein inclusions immunoreactive for several proteins but primarily comprising alpha-synuclein (α-syn) and α-syn phosphorylated at position S129 [6,7,8]. Neurons most vulnerable for LB pathology are projection neurons with disproportionately long axons and poor myelination—a key example being dopaminergic neurons projecting from the substantia nigra (SN) into the striatum [6,9,10]. Both genetic factors and environmental exposure build the basis for defining the causality of LBD [5,11,12,13,14,15]. A number of Mendelian forms of PD have been described in the last two decades in addition to risk genes identified through candidate gene studies and genome wide-association studies (GWAS) [16,17]. One of the largest GWAS studies to date reports 90 independent genome-wide significant risk signals across 78 genomic regions genotyping 7.8 million single nucleotide polymorphisms (SNPs) in 37,700 cases, 18,600 proxy cases, and 1.4 million controls [18]. Interestingly, tissue-specific expression enrichment analyses suggest that PD loci are heavily brain enriched, e.g., in neurons from the SN, globus pallidus (GP), thalamus, posterior cortex (PC), frontal cortex (FC), hippocampus (HC), and entopeduncular nucleus (ENT, internal part of the globus pallidus) [18]. Disease mechanisms and phenotypes related to PD range from toxic protein aggregation, impairment of mitochondrial and lysosomal/autophagy function, disruption of vesicle and endocytosis, and dopamine metabolism. Head injury and lifestyle factors as well as exposure to environmental toxicants and contaminants, including pesticides, solvents, or metals, among other pollutants, have been found to increase risk for PD and have been reproduced and validated in animal models [19,20,21].

Autism spectrum disorder (ASD), on the other hand, is a multifactorial neurodevelopmental disorder with a prevalence of 1.0–2.6%, presenting with three core symptoms: Impairments in social interaction; communication impairments; and restricted, repetitive, and stereotyped patterns of behaviors [22,23]. Additional features that can accompany ASD are motor abnormalities, gastrointestinal problems, epilepsy, intellectual disability, or sleep disorders [24]. Males are more frequently diagnosed with ASD than females, with a prevalence ratio of about 4:1 (male:female) [25,26]. Key neuroanatomical features of ASD are early brain overgrowth during toddler years, which disappears between 5–6 years of age [27]. Regions affected include the fronto-temporal and frontoparietal regions, the amygdala-hippocampal complex, cerebellum, basal ganglia, and anterior and posterior cingulate regions [28]. Microscopically, there are a decreased perikaryal size in layers III and V of the fusiform gyrus of the cortex, altered cell distribution of cortical layers with a less distinct lamination in the architecture, and abnormal neuronal morphology of von Economo neurons (VENs) [29] in the fronto-insular cortex and anterior limbic area with corkscrew dendrites and swollen soma [30]. The time period of early brain overgrowth suggests that neurodevelopmental processes are affected, including synaptogenesis, axonal and synaptic pruning, and myelination, leading to an altered ultrastructure of synapses and structural connectivity [31]. Clinical genetic studies identified more than 100 ASD candidate genes with mostly rare cytogenetic aberrations or single-gene mutations in 10–25% of cases [26]. Copy number variants (CNVs) are reported in 44% of familial ASD cases and ~7–10% of sporadic cases [32]. However, only a small number of genes have been proven causative at this point and genetic testing results need to be reviewed critically in the context of the mutation, gene function, gene modifiers, and potential protein interaction networks, gene environment interactions, or sex-linked modifiers [33,34]. These causative genetic factors converge on mechanisms primarily related to neuronal development, plasticity, synaptic structure, and performance [35,36], e.g., neurexin (*NRXN),* neuroligin *(NLGN)*, SH and multiple ankyrin repeat domains (*SHANK)*, tuberous sclerosis 1/2 protein (*TSC1/2)*, FMRP translational regulator 1 (*FMR1)*, or methyl-CpG-binding protein 2 (*MECP2)*, which are related to synaptic performance [25,26,35]. In addition, environmental factors have a significant contribution to disease risk of ASD as established in twins [37] and include prenatal viral infection, zinc deficiency, abnormal melatonin synthesis, maternal diabetes, parental age, and other potential postnatal risk and stress factors [38].

## 2. Overlapping Clinical Motor and Behavioral Phenomenology between ASD and PD

Loss of dopaminergic signaling in the nigro-striatal pathway is the primary cause for the motor symptoms in PD, causing an imbalance in the neuronal circuits of basal ganglia. In addition, it is becoming increasingly clear that the balance of the basal ganglia circuit is also altered in ASD, affecting both cognition and motor function [39,40]. Furthermore, it has been postulated that impairment in cortical control of striatal neuronal circuits underlies impulsive and compulsive activity—symptoms in both PD and ASD [41].

There are overlapping clinical phenomena in the cognitive and behavioral profiles of ASD and PD that can present independently of secondary causes, such as medication, e.g., neuroleptics or dopamine agonists [42]. In a clinical research study in two cohorts of ASD with an age over 39 years, there was a surprising increase in Parkinsonian motor features in 20% of cases (without neuroleptics use), and 7% of participants in this cohort were also clinically diagnosed with PD prior to the study [43]. Non-motor features are also commonly shared between ASD and PD, demonstrated by a higher prevalence of depression and anxiety, and obsessive–compulsive disorders (OCDs) [2,44]. Patients with ASD show a prevalence of anxiety in 39.6% [45], depression in ~30% [46,47,48,49], and OCD in 17.4% of cases. In PD, the prevalence of anxiety is described in 65–68.42% [50,51], depression in 13.8–56% [50,51,52], OCD in 37.5% (drug naïve) [53], and impulse control behavior (ICB) in 17.1–28.6% [54,55] of patients with PD. ICB in PD is illustrated by gambling, compulsive eating, punding, excessive shopping, or hypersexuality. A recent study showed an improvement of ICB symptoms after changing medication from dopamine agonists to levodopa/carbidopa slow-release formulations [56].

There are also rare cases of co-morbidity of ASD and early onset PD. One clinical case report describes a patient, diagnosed with PD at the age of 30, with left-side-predominant hypokinetic-rigid syndrome. By history, the patient had delayed language and motor development. During childhood, he had problems with social contact and showing emotions. In a formal assessment for ASD, the patient scored positive on all three domains (deficits in social interaction, communication, and stereotypical repetitive behavior), but genetic analysis was not performed [57].

A study in 39 children with ASD (age 2–8 years) without known karyotypic abnormalities showed a significant 3-fold decrease in α-syn plasma levels versus controls, which links one of the key proteins in PD pathogenesis to ASD. In this study, beta-synuclein (β-syn) levels were also found to be significantly higher in ASD children compared to the control group, postulating a compensatory effect of β-syn [58]. This overlapping clinical phenomenology of motor and non-motor symptoms and the molecular link between ASD and PD point towards potentially converging disease mechanisms and imbalances of dopamine signaling in the basal ganglia.

In summary, we describe striking overlapping clinical symptoms of Parkinsonian motor problems in ASD and psychiatric symptoms, such as depression, anxiety, and impulsive and compulsive activity, in ASD and PD. Besides genetic causes, it will be important to assess the overlapping molecular pathology of ASD and PD, including the inflammatory response and mitochondrial dysfunction, that might even serve as early molecular biomarkers. In clinical practice, the commonality of these two diseases points towards the importance of an interdisciplinary clinical care team comprising of movement disorder specialists and psychiatrists for the clinical care and treatment of ASD and PD. This would allow not only for patients to be served better but also for insight to be gained into the common disease etiology and advancing the development of novel molecular treatments.

## 3. Genes Linked to Neurodegenerative Diseases Also Play a Role in Neurodevelopmental Disorders

Genetic linkage studies and GWAS have provided deep molecular insight into disease mechanisms for both PD and ASD. While in PD, point mutations and non-coding risk variants have been predominantly described, only CNV alterations have been described in ASD; however, in the next section, we describe the frequencies for CNVs in the *PARK2* gene and 22q11.2 deletions, and FMR1 repeat expansions, which can present both with clinical symptoms of PD and ASD. However, α-syn CNVs play a major role and comprise about 50% of reported mutations [32]. In the gene (*SNCA)*, duplications and triplications have been identified as a rare cause of PD, although genomic *SNCA* deletions have not been described in the literature. The mining of public genomic databases, however, revealed cases heterozygous for genomic *SNCA* deletions, which also present with developmental delay and ASD [59] (Appendix A).

### 3.1. PARK2 Copy Number Variants in PD and ASD and its Substrate GPR37 Are Linked to Autism

Point mutations and copy number variants in the *PARK2* gene (PRKN, parkin RBR E3 ubiquitin protein ligase, OMIM *****602544) have been implicated in autosomal recessive juvenile Parkinsonism first described in 1998 in Japanese families and is the most common genetic cause in early onset PD [60,61]. The *PARK2* gene on chromosome 6q26 is the second-largest gene in the human genome and a common fragile site (FRA6E) in the human genome. *PARK2* point mutations and CNVs have also been described in glioblastoma, ovarian, and liver cancers [62,63,64,65].

Strikingly, it has not been widely recognized that CNVs encompassing the *PARK2* gene are also found in cohorts of patients with ASD/developmental delay and attention deficit hyperactivity disorder (ADHD) at a frequency of 0.5–2.2%. There are single clinical case reports of deletions and duplications of the *PARK2* gene [66,67], but GWAS studies and clinical diagnostic case series also support these findings. The first whole-genome CNV study in 2009 detected *PARK2* deletions in seven ASD cases (0.8%) and none in controls among 859 ASD cases and 1409 controls of European ancestry [68]. In a more recent CNV GWAS study in 335 ASD cases and 1093 healthy controls in Han Chinese, 6 CNVs (1.8%) were detected in the *PARK2* region in the ASD cohort (four deletions and two duplications in exons 2–7, with a CNV size between 50 and 250kb), while only two duplications were found in controls (0.2%) [69]. Furthermore, in an ADHD cohort of 875 young patients and 2066 matched controls (test and replication groups combined), 16 *PARK2* CNVs (1.8%) were identified in the ADHD cohort and 3 (0.1%) in the controls [70]. A diagnostic case series from Czech Republic describes two CNVs (2.2%) in the *PARK2* gene, one deletion and one duplication out of 92 ASD cases [71]. A second consecutive diagnostic case series from the University Kansas included 215 patients referred for autism/ASD or developmental delay/learning disability and found one *PARK2* CNV duplication (0.5%) in a patient with learning disability and dysmorphic features [72].

In the Autism Genome Project (AGP) cohort (European and Northern American ancestry), 15 out of 2446 ASD cases (0.61%) harbored partial exonic *PARK2* CNV deletions or duplications [73] (Figure 1A, Appendix A). We excluded 31 CNVs in this region that did not encompass exons and, hence, we could not establish causality according to the American College of Medical Genetics (ACMG) guidelines [74]. Of the 15 *PARK2* CNVs included, 6 were deletions and 9 were duplications, spanning coding exons similar to CNV mutations found in juvenile PD [61].

The *PARK2* gene encodes parkin, an E3 ubiquitin-protein ligase, which is recruited to damaged mitochondria to ubiquitylate outer membrane proteins to promote clearance through mitophagy and autophagic pathways [76]. Parkin was also been found to interact with pyruvate kinase isozymes M2 (PKM2), regulating the glycolysis pathway [77]. Parkin has a number of substrates, such as mitochondrial proteins and endosomal trafficking regulators recently described in a drosophila screen, including vacuolar protein sorting-associated protein 35 and 4 (VPS35, VPS4), arginine kinase (ArgK), and reductive dehalogenase anchoring protein (RdhB) [78,79]. Relevant to neurodegeneration, Parkin has been reported to interact with a glycosylated form of α-syn (a-Sp22) [80]; synphilin, an α-synuclein-interacting protein [81]; and synaptotagmin XI (SYT11), involved in regulation of the synaptic vesicle pool and release [82,83,84]. In addition, parkin ubiquitinates the parkin-associated endothelin-receptor-like receptor (Pael-R/GPR37) and promotes degradation of its insoluble form [85]. Pael-R/GPR37 interacts with the dopamine transporter (DAT, SLC6A3) and modulates DAT activity [86]. Pael-R knockout mice show an increase in DAT expression and resistance to methyl-phenyl-tetrahydropyridine (MPTP) exposure. Loss of parkin function leads to accumulation of Pael-R/GPR37, which is described in post-mortem brains of *PARK2* cases. Pael-R/GPR37 is also found in the core of LBs and Lewy neurites [87]. During neuronal development, GPR37 is required for Wnt-dependent neurogenesis and functions in the maturation of the N-terminal bulky b-propellers of the Wnt co-receptor low-density lipoprotein receptor (LDLR)-related protein 6 (LRP6) [88].

To date, no mutations or risk variants in the *Pael-R*/*GPR37* gene have been described in PD; however, several mutations in the *GPR37* gene (chr7q31–33, *AUTS1* region) have been described in ASD patients. A c.1585–1587del TTC (Del312F) in a patient of Japanese descent, c.2324G > A (R558Q) in one Caucasian patient, and T589M was found in seven affected Caucasian patients [89]. Causative *GPR37* mutations could lead to a loss of function, resulting in an increase of DAT expression and activity.

### 3.2. 22q11.2 Deletion/Di George Syndrome Linked to Parkinsonism

While *PARK2* is a bona fide PD gene, deletions of chromosome 22q11.2 (22q11.2DS, OMIM **#** 188400) have only recently been implicated in PD [90,91]. The 22q11.2 deletion is considered to be one of the most common deletion syndromes (~1:1000–3000 births) and was first described in 1968 [92,93]. The majority of 22q11.2DS encompass a 3-Mb region including ~90 genes, whereas 5–10% of patients have a smaller 1.5-Mb deletion. About half of these genes are expressed in the human brain [94]. The 22q11.2 deletion is a heterogeneous clinical syndrome involving neurodevelopmental, psychiatric, cardiac, immunological, and endocrinological abnormalities. Children with 22q11.2DS have developmental delay and are at a higher risk for ASD, ADHD, and other mental disorders, including depression and anxiety [95]. The complex clinical presentation is due to the large number of genes in the region. Cases with smaller atypical CNVs in this region contribute to the understanding of the underlying causative genes, i.e., the critical region for a higher rate of autism encompasses the low copy repeat (LCR) region A to B and increases the risk of autism by ~40% [75] (Figure 1B, Appendix A). LCRs can mediate chromosomal rearrangements and there are a total of eight LCRs within 22q11 [96,97]. In the AGP project, 59 CNV deletion cases (ranging from 6.2 kb to 2.6 Mb) were detected within the 22q11.2 region in an ASD cohort (2.4%) (Figure 1B, Appendix A). Ten cases had a deletion surrounding the LCR-A to LCR-B region [75]. Several smaller deletions in the region could pinpoint putative candidate genes relevant in the context of ASD.

Recently, it has been recognized that adults with 22q11.2DS have a higher risk of developing PD [91] as illustrated by a PD prevalence of 5.9% in a cohort of 159 adults (ages 35–64 years) with 22q11.2DS [98]. Clinically, the onset of symptoms is asymmetric, with typical progressive motor features of bradykinesia, rigidity, and tremor. Patients respond well to levodopa therapy. PD onset in 22q11.2DS cases is often earlier, with a mean age 39.5 ± 8.5 years (range, 18–58) [99]. Other symptoms include early dystonia, a history of seizures, and neuropsychiatric symptoms (e.g., psychosis and anxiety) [100].

A mouse model encompassing the 22q11 genomic deletion region, the Df1/+ model [101], was found to have elevated levels of α-syn and p62, a key receptor for autophagy. They showed behavioral motor coordination deficits, which could be reversed by pharmacological inhibition of mammalian target of rapamycin (mTOR) activity by a rapamycin analog [102]. While it is difficult to pinpoint causative genes in this large deletion region implicated in PD or ASD, several genes would qualify as candidate genes., e.g., genes related to mitochondrial dysfunction, namely *MRPL40, PRODH, SLC25A1, TANGO2, TXNRD2,* and *ZDDHC8*. In this regard, disruption of a network of inner mitochondrial membrane transporters (SLC25A1-SLC25A4) required for synapse function has been identified in the Df1/+ mouse model and patient biomaterial from 22q11.2DS cases [103]. Another potential candidate gene is the catechol-*O*-methyltransferase (*COMT*) gene, which degrades catecholamines (i.e., dopamine or norepinephrine). Lower levels of COMT would lead to higher synaptic dopamine levels following neurotransmitter release, ultimately increasing dopaminergic stimulation at postsynaptic neurons and causing dopamine toxicity.

### 3.3. FMR1 Intermediate CGG Repeat Expansion Causes Autism and FXTAS/Parkinsonism

Variable CGG repeat expansions in the 5′ untranslated portion of the fragile mental retardation 1 gene (*FMR1,* chrXq27.3) cause a spectrum of disorders, including fragile X syndrome (FXS, OMIM **#** 300624) (>200 CGG repeats), fragile X-associated tremor ataxia syndrome (FXTAS, OMIM **#** 300623; permutation 55–200 CGG repeats), and fragile X-associated primary ovarian insufficiency (FXPOI, premutation) [104].

Both the full repeat expansion and the premutation give rise to autism/ASD. It is estimated that 30% of FXS cases have autism and 2–6% of all cases with autism carry FMR1 repeat expansions. In addition, premutation carriers can develop FXTAS, a neurodegenerative disorder characterized by cerebellar ataxia, intention tremor, and cognitive impairment usually starting >65years of age. Interestingly, some patients with FMR1 premutations present with symptoms compatible with typical PD [105,106,107]. LB pathology has been described in FXTAS cases [108] but also tau pathology either consistent with Alzheimer’s disease or progressive supranuclear palsy (PSP) [109,110,111,112]. While the full FMR1 repeat expansion leads to deficiency or absence of the *FMR1* protein (FMRP), the FMR1 premutation leads to RNA toxicity and results in eosinophilic intranuclear FMR1-mRNA-containing inclusions in neurons and astrocytes in various brain regions, including the cortex, basal ganglia, thalamus, hippocampus, amygdala, and SN [109]. The FMR1 protein functions as an RNA carrier protein controlling the translation of genes regulating synaptic development and plasticity.

### 3.4. SNCA Genomic Region (4q22.1) Is a Multiplication/Deletion Syndrome for PD and ASD

α-syn (encoded by *SNCA*, NM_000345.3, OMIM *163890) is a key protein in the pathogenesis of PD and related α-synucleinopathies [113]. Since its discovery as the first PD-causing gene in 1997 [114,115,116], several missense mutations, a number of risk SNPs, small structural variants (SSVs), and, recently, cases with *SNCA* multiplications and somatic mosaicism have been described [59,117,118]. Genomic duplications and triplications of various genomic sizes of the 4q22.1 genomic region are highly penetrant and contribute to PD, LBD, or multiple system atrophy (MSA). The critical genomic region of these cases spans the *SNCA* gene, which is the only gene overlapping in all described cases [59,118,119,120]. While earlier studies used single PCR-based copy number analysis, comparative genomic hybridization (CGH) and optical mapping can now define the size of CNVs, breakpoints, and orientation much more accurately [121] (Appendix A).

The underlying mechanism for these pathogenic duplications or triplications is intrachromosomal non-allelic homologous recombination (NAHR), which is a misalignment of homologous chromosomes during meiosis leading to the recombination and exchange of different regions of the chromosome, resulting in alterations of the genomic structure, such as duplications or deletions [122,123].

While multiplications of the *SNCA* locus are causative for neurodegeneration and LB pathology, deletions of this region on chromosome 4q22.1 have not been explicitly described as disease causing. In general, it is thought that lower levels of α-syn are possibly neuroprotective, hence, individuals with *SNCA* deletions would not contract disease. However, a closer look into public disease databases (e.g., ClinVar, Decipher, and ISCA) and large cohort studies for CNV variation in neurocognitive disorders reveals that there are patients with large deletions of the *SNCA* locus, which clinically present with developmental delay, ASD, and/or congenital abnormalities. Of note, these cases are distinct from chromosome 4q deletion syndromes, which encompass either a proximal deletion between 4(q11–q31) or distal deletion of 4(q31–q35) [124,125].

Fifteen *SNCA* CNV cases are described in the ClinVar and Decipher databases, ranging from 67 kb to 9 Mb, with the smallest overlapping region as the *SNCA* gene (Figure 1C,D, Appendix A). These are cases submitted through clinical diagnostic labs and detailed clinical information is not available for these patients. In addition, there are several large cohort studies with children with developmental delay, intellectual disability, and/or ASD, in which cases with SNCA CNV deletions are described (Figure 1C,D, Appendix A).

The Decipher CNV database lists three cases in total. Two of them have large deletions 5.75Mb (ID 337659, NCBI36/hg18; chr4:89571443-95328195) and 6.25Mb (ID 339430, NCBI36/hg18; chr4:88295927-94547516) with multiple genes affected, and a third case (ID 251304, NCBI36/hg18; chr 4:90527679-90705624) carrying a duplication of 177,946 bp partially spanning the SNCA gene only. This partial *SNCA* duplication is predicted to result in a loss of function of α-syn. The clinical presentation of this case is broadly described as having facial dysmorphology and intellectual disability. Another duplication was described in the AGP cohort upstream of the SNCA gene. While the pathogenicity is unclear, this CNV resides within a non-coding region of evolutionarily conserved elements that have been shown to modulate gene expression in cellular reporter assays (Figure 1D) [126].

Vulto-van Silfout et al., 2013 described a cohort of 5531 children, ages 3–17, with intellectual disabilities, ASD, and/or congenital abnormalities. They found CNVs in 25.1% of the cases using a 250,000 single-nucleotide polymorphism array platform. One patient carried an 884-kb deletion within the *SNCA* genomic region encompassing *SNCA, MMRN1*, and an unknown gene *KIAA1680* (NCBI36/hg18; chr.4:90,629,298-91,514,095) [127]. Coe et al., 2014 described a large CNV analysis of 29,085 children with developmental delay and/or ASD in comparison to 19,584 healthy controls using array comparative genomic hybridization (CGH). Seven patients and two controls carried a 475-kb deletion (NCBI36/hg18; chr.4:90,793,560-91,268,616), including only the *SNCA* and *MMRN1* genes [128].

### 3.5. Four PD-Associated Genes Are Linked with Neurodevelopmental Disorders

In summary, we have described genetic conditions that can give rise to ASD or PD/LBD depending on their gene dosage, establishing precedence that there are potential converging mechanisms and pathways that play a role both in neurodevelopment and neurodegenerative processes. Partial *PARK2* deletions/duplications cause early onset PD when they lead to complete loss of protein in compound heterozygote or homozygote individuals, whereas heterozygote carriers are found in autism populations, which still express 50% of the parkin protein, but in these cases lead to haploinsufficiency. On the contrary, expansions of the FMR1 gene >200 CGG repeats lead to a loss of FMRP protein and autism, but the premutation causes PD, with the RNA found in foci in the brain of FXTAS cases [109]. The 22q11 deletion can lead to both ASD and PD as a continuous symptom spectrum similar to cases with trisomy 21 and the development of Alzheimer’s disease [129,130]. Lastly, deletions and partial duplications of SNCA gene pinpoint to a new critical region for developmental delay and ASD on chromosome 4q22.1. We propose a new model for the *SNCA* multiplication/deletion region. While overexpression of the SNCA gene has been accepted as being causative for PD and related neurodegenerative conditions, we propose that deletions or partial duplications of the SNCA genomic regions lead to developmental delay and ASD and relevant models could shed light on this connection and mechanisms possible linked through synaptic function and neuronal development.

## 4. Physiological Function of α-syn

α-syn is abundantly expressed throughout the brain [131,132] and localizes in higher concentrations in the presynaptic nerve terminals [133,134]. The conformational plasticity of α-syn [135,136], and its ability to interact with a myriad of partners and biological membranes [137,138,139] explain the multifunctional properties of α-syn in the cell and, in particular, at the synapse (Figure 2).

α-syn is an important mediator of synaptic homeostasis and neurotransmitter release, regulating synaptic vesicle mobility and pool size [140,141,142,143,144]. One of its physiological roles is to bind to and induce the clustering of synaptic vesicles as described in in vitro and in vivo models (Figure 2) [145,146,147]. In addition, α-syn binds to vesicle-associated membrane protein 2 (VAMP2, synaptobrevin-2) and functions as a chaperone for the Soluble N-ethylmaleimide sensitive factor attachment protein receptor proteins (SNARE) protein complex, which regulates the docking of synaptic vesicles and assists their assembly and distribution [147,148,149]. Recent studies provide further evidence for the physiological function of α-syn in maintaining synaptic vesicle clustering, hence attenuating synaptic vesicle recycling and neurotransmitter release [150]. The interaction with VAMP2 seems to be essential, as engineered α-syn lacking VAMP2 binding sites failed to induce vesicle clustering [150]. Besides VAMP2, α-syn interacts with a number of other presynaptic proteins, which are known modulators of synaptic vesicle clustering and recycling, like the synapsins [151]. The interaction of α-syn with synapsins facilitates its function and helps regulate vesicle mobility by influencing α-syn targeting to synaptic vesicles [151,152,153,154]. In particular, synapsin III has been shown to have an effect on clustering of the reserve vesicle pool. In the absence of α-syn, there is an increase of synapsin III in synaptic terminals (Figure 2), which alters the proper clustering of synaptic vesicles at the active zone and leads to a reduction of dopamine release [154].

It has been reported that elevated or mutant α-syn leads to a reduction in dopamine release possibly by affecting exocytosis or decreasing vesicle availability in the recycling pool due to impaired vesicle endocytosis (Figure 2C) [155]. In contrast, decreased levels of α-syn reduces the availability of vesicles in the reserve pool and consequently more vesicles are readily available to be released, leading to an increase in dopamine release (Figure 2A) [155]. The role of α-syn in vesicle trafficking is also supported by its connection to cytoskeleton protein networks. Specifically, α-syn binds subunits of tubulin and influences microtubule organization and dynamics [156,157]. Furthermore, α-syn can modulate actin microfilament dynamics, thus affecting the plasticity of the cytoskeleton involved in synaptic vesicle mobility and their organization at the active zone [158].

Synaptic signaling in nerve terminals is tuned by monoamine transporters that govern the reuptake of their associated neurotransmitters and strictly regulate dopamine, serotonin, and norepinephrine extracellular concentrations. In cooperation with the microtubule network, α-syn influences the trafficking, activity, and surface localization of the monoamine transporters at the plasma membrane [159,160,161]. Indeed, the absence of α-syn can impair dopamine transporter (DAT) function and increase extracellular dopamine levels (Figure 2A) [162]. Moreover, a reduction in dopamine reuptake and impaired DAT function in the dopaminergic neurons is observed in animals overexpressing wild-type α-syn (Figure 2C) [163]. Another key regulator of dopamine signaling is the availability of vesicular monoamine transporter (VMAT2) on synaptic vesicles. VMAT2 transports dopamine and regulates its cytosolic levels. In this context, increased levels of α-syn could decrease VMAT2 activity (Figure 2C, bottom panel) and potentially affect dopamine homeostasis [164]. Taken together, studies investigating the physiological functions of α-syn posit that it is essential for normal synaptic performance, which is further detailed in rodent synuclein knockouts models.

## 5. α-syn Knockout Mouse Models

Several knockout (KO) models of synucleins have been generated to better understand the physiological function of α-syn. A single knockout of the murine α-syn gene missing amino acids 1–41 of the protein [165], a spontaneous *Snca* KO in tandem with deletion of multimerin 1 (Mmrn1) C57Bl/6S strain [166,167], a double KO of α-syn and β-syn [168], and other combinations [169,170] as well as a triple KO model of all synucleins have been created [148,171,172] (Appendix A).

In this context, it is important to know that α-syn belongs to a family of structurally related proteins that include three members: α-syn, beta-synuclein (β-syn, chromosome 5q35; OMIM 602569) [173], and gamma-synuclein (γ-syn, persyn, chromosome 10q23.2-q23.3; OMIM 602998) [174]. All synucleins have a highly conserved apolipoprotein-like class-A2 helix repeat domain and share a common natively unfolded tertiary structure [134]. Besides high levels of α-syn in peripheral red blood cells [175], α-syn and β-syn are predominantly expressed in the cortex, hippocampus, striatum, and ventral midbrain and largely co-localized in presynaptic nerve terminals. While γ-syn is expressed in the peripheral and central nervous system, it is also found in ovarian tumors and olfactory epithelium [134,176,177,178].

### 5.1. α-Syn KO Model Accelerates the Recovery of Synaptic Vesicle Release

*α-syn* knockout mice (α-Syn-KO, background strain C57/BL6 F2) are viable, normal in size, and fertile without gross anatomical abnormalities. They display normal expression of β-syn and γ-syn [165]. Dopaminergic neurons of α-Syn-KO mice appear morphologically indistinguishable from those from wild-type mice and the number of tyrosine hydroxylase (TH)-positive neurons in the midbrain is not changed. The density of dopaminergic projections to the striatum is normal by TH immunohistochemistry (IHC), but a reduction of dopamine content by 18% in the striatum but not in the ventral midbrain is reported. Dopamine metabolite levels of 3,4-dihydroxyphenylacetic acid (DOPAC) are unaltered [165]. In the same model, a small (17.8%) but statistically significant decrease of TH-positive neurons in the SN pars compacta, but not in the ventral tegmental area (VTA), is detected in nine-month-old animals. Furthermore, the striatal DOPAC to dopamine ratio remains unchanged [179]. Old α-Syn-KO mice (24–26 months of age) consistently show reductions of dopamine, TH-positive fibers, and DAT in the striatum [170], indicating that loss of α-syn leads to a small decrease in dopamine neurons in the SN over the lifespan of an organism. The C57Bl/6S mice that carry the spontaneous *Snca/Mmrn1* deletion show a 33% decrease in developing dopaminergic neurons at embryonic day (E) 13.5, which results in a lower count of TH-positive neurons postnatally [180].

In synaptic terminals of α-Syn-KO mice, the protein composition along with their density and structure remain unchanged, as shown by IHC and electron microscopy of the presynaptic terminal proteins synaptophysin, Rab3a, and β-syn, indicating that α-syn may not be a necessary structural component of presynaptic terminals [165]. Interestingly, α-syn deficiency in the α-Syn-KO model accelerates the recovery of dopamine vesicle release in the striatum by paired pulse electric stimuli, suggesting an inhibitory role of α-syn in activity-dependent modulation of vesicle release, whereas amphetamine-stimulated DA release was not different. In another study, a deficiency of undocked synaptic vesicles in the hippocampus was found in α-Syn-KO mice (129/SvEvTac background), which showed significant impairments in the synaptic response to a prolonged train of repetitive stimulation, with a consequent delay of the replenishment of docked vesicles from the reserve pool, indicating that α-syn regulates synaptic vesicle release [181].

### 5.2. Double Synuclein KO Models Show Compensatory Increase of β- or γ-Syn

A confounding factor in studying the physiological role of α-syn is the presence of β- and γ-syn, because these highly homologous isoforms are thought to be able to functionally compensate for each other at least in part. Therefore, double and triple synuclein KO mouse models would be able to account for this functional overlap (Appendix A).

Double α-β-Syn KO mice do not show impairment of basic neuronal function or survival but present with a reduced concentration of 18–25% striatal dopamine [168,170]. Compared to single α-Syn-KO mice, double α-β-Syn KOs show a similar phenotype, but with several key distinctions: (1) γ-syn was increased by 50% presumably as a compensatory effect; (2) conserved regulatory binding proteins 14-3-3ε protein were increased by 30%; (3) 14-3-3ζ protein was decreased by 30%; and (4) complexines, cytoplasmic proteins that bind to SNARE proteins, were increased by 30%. Analysis of the synaptic terminals revealed that the vesicle number and size remained unchanged [168]. In this context, it is relevant to point out that 14-3-3 proteins interact with α-syn [182] and are also found in LBs [183]. In addition, 14-3-3θ has been found to be decreased in human wild-type α-Syn transgenic mice [184]. While 14-3-3 and α-syn can both regulate TH, 14-3-3 can interact with phosphorylated TH to promote dopamine synthesis and α-syn may bind to dephosphorylated TH to counteract dopamine synthesis and reduce TH activity [185].

In addition, α-γ-Syn double KO mice were generated by cross-breeding γ-Syn C57B16J null mice [186] with the α-Syn-KO mice [165]. These α-γ-Syn-KO mice exhibit a 15–20% decrease in TH-positive neurons in the SNpc while DOPAC, 5-hydroxytryptamine (5-HT), and homovanillic acid (HVA) levels remain unchanged [179]. The model also displayed an increased concentration of β-syn [179]. This increase of β- or γ-syn suggests a partial functional overlap.

### 5.3. Triple SYN-KO Models Display Smaller Synapses

To completely rule out functional overlap of synucleins, mouse models with deletions of all synuclein family members have been generated by cross-breeding single or double SYN-KO models [148,170,171,172]. Young triple SYN-KO mice displayed no obvious phenotype but developed severe neurological impairments with age and a reduced life span (death by 16 months of age). These mice had an age-dependent decrease of VAMP2/synaptobrevin-2, increase in cysteine-string protein alpha (CSPα), and a decrease in SNARE complex proteins. These results indicate that synucleins are required for maintaining normal SNARE-complex assembly [148].

The number of TH-positive neurons in the SNpc of triple KO mice showed a small, but not significant, reduction and did not change with age [170,172,179]. Triple SYN-KO mice (at 3, 12, and 24 months) had a normal overall brain architecture and morphological architecture. In the triple Syn-KO model, a reduced dopamine content was reported in the dorsal striatum [170], similar to the single α-Syn-KO mice [165,187]. However, the synaptic puncta of the CA3 hippocampal neurons in triple-Syn-KO mice were around 30% smaller compared to animals at three months of age. This demonstrates that deletion of synuclein isoforms has a direct effect on synapse structure [171]. Analyzing the changes in synaptic composition, 3-month-old triple-Syn-KO mice showed an increase in complexin II, synapsin IIb, and 14-3-3β and 14-3-3ε isoforms [171], comparable to the results in the double α-β-Syn-KO mice [168].

### 5.4. Syn-KOs Have Learning and Memory Deficits, Hyperdopaminergic-Like Behavior, and Resistance to MPTP

In behavior studies, α-Syn-KO mice (6–10 months) had significantly fewer rearing responses and a lower ratio of center to total distance in the open field test, which can be interpreted as an anxiety-related phenotype [181]. Furthermore, 10-month-old α-Syn-KO mice have a significantly reduced ability to learn and a deficit in working and spatial memory compared to wild-type littermates [188,189]. α-β-Syn KO mice show a decrease of motor performance [170]. Other studies did not find changes in motor coordination [165,187]. Mice exhibit a hyperactive behavior in novel environments and a hyperdopaminergic-like behavior with age [172].

A resistance of nigral dopaminergic neurons to MPTP toxicity was observed in the single α-Syn, γ-Syn, as well as in double α-γ-Syn, and α-β-Syn KO models [179,190,191,192].

### 5.5. Overall Findings of α-Syn Murine Models

α-syn knockout and overexpression models have been instrumental for addressing the physiological functions of α-syn. In addition to reduced SNARE-complex assembly and associated proteins, and changes in 14-3-3 proteins [148], Syn-KO mice exhibit changes in synaptic structure and size [171], resulting in cognitive and memory impairments [188,189]. Additional evidence for the role of α-syn in synaptic vesicle trafficking and maintenance is illustrated by a slower refilling of the docked vesicle pools from the reserve pool in mice lacking α-syn [165,181]. Although single gene α-Syn KO models only show mild synaptic phenotypes, possibly due to compensatory mechanisms from other synuclein isoforms, under conditions capable of exhausting docked vesicles as well as reserve vesicle pools, significant impairment of synaptic responses become evident [181]. All single, double, and triple Syn-KO models showed a reduction in striatal dopamine concentrations and unchanged dopamine metabolites, such as DOPAC, HVA, and 5-HT (a dopamine-specific effect as serotonin levels are unaffected), indicating that the dopamine regulation was affected by either the capacity of dopamine storage or release from vesicular stores [168,170,179,187].

Interestingly, some of these findings are also observed upon modest α-syn overexpression, where α-syn interferes with synaptic vesicle recycling, inhibit exocytosis, and leads to impairments in neurotransmitter release [193,194,195]. Interestingly, overexpression of wild-type α-syn, but not the A30P mutant, suppresses dopamine release [196], implying that the affinity of the α-syn protein for membranes plays an important role in the observed presynaptic disturbance [196].

## 6. α-Syn Expression in the Developing Brain

The number of nigral neurons declines during normal aging but at an accelerated rate in PD, and motor signs of PD only develop after the degenerative process has reached a certain threshold, with a 60% decline in striatal dopamine [197]. Compensatory mechanisms can mitigate the beginning of the symptoms. It is feasible that individuals with a genetic form of PD are born with a smaller number of nigral neurons or a different morphology that makes them more susceptible to neuronal degeneration [198]. It is known that the accumulation of α-syn into higher molecular weight aggregates triggers cellular and molecular dysfunction, such as autophagic/lysosomal, endoplasmic reticulum, and mitochondrial dysfunction, leading to neuronal cell death and PD [198,199,200,201,202].

### 6.1. α-Syn in Murine Development

The generation of all neuronal populations in the brain includes proliferation and migration processes of precursors starting with the neural plate folding into the neural groove and neural tube [203]. In mice, after the neural tube closes, neuronal generation and migration start around E9 (Figure 3B, green line) [204]. Since α-syn is expressed early in development, it is likely that α-syn expression has an important role in the temporal and spatial development of neurons. α-syn is first detected as early as mouse embryonic day 9.5 (E9.5) (Figure 3B, red line) and is restricted to the midbrain/hindbrain junction [205]. At E10.5, α-syn is expressed in the olfactory bulb, neocortex, hippocampus, dorsal root ganglia (DRG), globus pallidus, nucleus accumbens, midbrain (including SN and VTA), hindbrain, and some regions of the marginal zone of the forebrain. At E12.5, α-syn expression is prominent in the spinal cord, sensory ganglia, and olfactory epithelium as well as in the forebrain and dorsal midbrain, most prominently in the deep cerebellar nuclei. At E14.5, expression in the dorsal midbrain starts to diffuse to the intermediate zone and expression in the telencephalon concentrates in the marginal zone and subplate. At E16.5, α-syn is expressed from the cerebellum to cortical regions, whereas expression in the deep cerebellar nuclei is weak. At the end of E18.5, the α-syn pattern is concentrated mostly in the subplate and it is less abundant in the cell bodies and more overt in axons [205]. Furthermore, the subcellular distribution changes over time from E9.5 to E14.5. α-syn is first expressed in the nuclei and cytoplasm, whereas after E16.5, α-syn is mostly distributed in the neuropil of neurons (Figure 3A) [205]. In the postnatal stage, α-syn mRNA expression is observed in the hippocampus and neocortex on postnatal day 1 (P1) and peaks at P7 [206,207]. In the ventral midbrain, α-syn mRNA expression is maintained into adulthood [208,209].

### 6.2. α-Syn in Human Neurodevelopment

Comparable to mice, dopaminergic neurons in the human brain encompass similar neurodevelopmental stages (Figure 3B, green line). First, the neural tube is closed by 3–4 weeks of gestation, which presents the foundation and precursor of the nervous system, then cells form the neural tube proliferate, creating the required neurons (neurogenesis) that will form all structures along the central nervous system [210,211]. Regions called “proliferative zones” form young neurons that will start the migration (Figure 3C, green line) process between weeks 12 and 20. These neurons migrate from their original zones to their target regions, moving along a scaffolding of glial cells [210,212]. The final step is to form connectivity along the different zones of the brain, synaptogenesis (Figure 3C, green dotted line), which begins around 27 weeks [210] and slowly establishes a dense multi-connecting wiring system. Synaptic density reaches its peak at 2 years of age and is followed by a specific loss and re-organization of synaptic connections, a process called pruning [211]. The human brain grows rapidly in the first 2 years (80% of adult weight); however, significant remodeling of grey and white matter continues into the third decade of life [213].

During human neuronal development, α-syn expression (Figure 3C, red line) occurs early, which suggests its importance in early stages of human development. Similarly to murine development, α-syn expression localizes mostly in the cell bodies of the immature neurons and then a subcellular re-localization to the axons and nerve fibers (neuropil) occurs later during neuronal maturation [179,205].

Very few studies have assessed α-syn in humans during neuronal development as tissues for such studies are scarce [214,215]. Brain specimens from 17 prenatal human fetuses from 11 to 39 post conception weeks (W) and 4 postnatal brains from 5 days to 16 years were analyzed for α-syn, β-syn, and γ-syn expression by Galvin et al., 2001 [215]. The approximate gestational age (post conception weeks, W) was determined by menstrual history, crown-rump length, and body weight. In this study, TH-positive neurons (a marker for dopaminergic neurons) were detected as clumps of round-shaped neurons as early as 11W, which migrated from the ventricular zone towards the developing SN. At 15W, more mature (pyramidal-like cells with short dendrites) TH-positive neurons were detected in the SN, and by 18–23W, dendrite-like processes became increasingly complex and started acquiring more mature morphological features defined by a multipolar shape with dendritic processes [215]. Interestingly, α-syn expression is detected as early as 15W in the forming SN. Then, as the DA continue maturing, redistribution of α-syn takes place (around the 18W), re-localizing from the cell body to the neuropil [215] (Figure 3A). This “re-localization” has also been reported for other synaptic vesicle proteins, such as synaptophysin, synaptotagmin, and synaptobrevin/VAMP2 [216]. These three synaptic vesicle proteins show a similar distribution compared to α-syn. At 15W, synaptophysin and synaptogamin are co-expressed in SN cell bodies and translocate later to the neuronal processes as mature neurons. In contrast, synaptobrevin expression starts later, at 18W, but re-localizes in the same way as α-syn, synaptobrevin, and synaptophysin [215]. Like α-syn, β-syn was detected at 15-17 W in the cytoplasm, and then at 19W, its expression was mostly in the neuropil. Finally, γ-syn expression was seen later, around 17W, as a perinuclear staining. Around 20–28 W, γ-syn expression was mostly localized in the cell body, and by 32W, it was also re-localized in the axons of the neurons [215].

In 2004, Raghavan et al., studied the distribution and temporal expression of α-syn in 39 brains (11W to 22 postnatal years (Y)) in different regions of the CNS. They especially described its expression in the neocortex and hippocampus, and included overall observations of the midbrain (SN, VTA) pons, medulla oblongata, etc. [214]. In this study, α-syn expression was first observed at 11W in the cell bodies followed by a general increase in several brain regions, with a peak at 28W (Figure 3C, red line). Comparable to Galvin et al., 2001, the same spatio-temporal re-localization of α-syn from the cell body of immature neurons to the processes of mature neurons was reported in most of the substructures analyzed. In contrast, some specific cell types, such as germinal matrix, glia, endothelial cells, external granular layer, Purkinje cells, and dentate neurons, remained consistently negative throughout development, indicating that α-syn is predominantly expressed in neurons [214].

### 6.3. α-Syn Overexpression or Lack of α-Syn Impairs Adult Neurogenesis

Adult neurogenesis of the brain occurs in the subventricular zone (SVZ), also termed the subependymal zone (SEZ), and dentate gyrus of the hippocampus and contributes to neuronal plasticity [217,218,219,220,221], although there is some recent controversy around human adult hippocampal neurogenesis [222,223,224]. In addition, there is evidence that ischemic trauma can stimulate neurogenesis in human and rodent models [225], and a recent study described neurogenesis in the adult striatum using carbon-14 dating and IHC for neuronal progenitors [226]. Interestingly, the SVZ receives dopaminergic afferents from the SN, serotoninergic fibers from raphe nuclei, and cholinergic axons from striatal neurons, and lesioning of the DA fibers lead to an impairment of progenitors within the SVZ [227,228].

Impaired neurogenesis has also been implicated as a key disease process in ASD among others, such as neurite growth, synaptogenesis, and synaptic plasticity [229]. Interestingly, several CNVs that predispose to ASD have been reported to impair neurogenesis, including 22q11.2 deletion syndrome, which we reviewed above as a risk factor for PD [230,231,232].

Overexpression of human α-syn results in reduced neurogenesis in the olfactory bulb granule and glomerular cell layer and in the dentate gyrus of the hippocampus of adult transgenic mice, as a result of diminished survival in neurogenic regions [233,234,235]. Aged transgenic mice for human wild-type α-syn and mutant A30P or A53T α-syn also show decreased neurogenesis caused by reduced cell proliferation in the SVZ and altered olfactory bulb (OB) neurogenesis [236,237]. Overexpression of human α-syn or A53T mutant α-syn leads to overt degeneration of postmitotic neurons in the dentate gyrus during postnatal development, indicating that these neuronal progenitors are more vulnerable to α-syn compared to mature neurons [238]. In addition to decreased neurogenesis, α-syn mutant oligomers of the E57K mutation lead to a severe reduction of synaptic mushroom spines and spine density [239].

To counteract α-syn-mediated defective adult neurogenesis, dopamine D2/D3 receptor activation through oral pramipexole treatment leads to an increase in adult neurogenesis in the SVZ–olfactory bulb system of a 6-hydroxydopamine (6-OHDA) model [240]. A follow-up study also concluded that the stimulation of hippocampal and dorsal striatal neurogenesis may be upregulated by pramipexole but not in the ventral striatum [241]. Overexpression of α-syn in the adult rat hippocampus leads to abnormal neuronal differentiation of NPCs mediated by a reduction of Notch1 expression and its downstream targets Hes1 and Hes5 [217], which play a critical role for dendrite development of immature neurons postnatally [242,243].

Additionally, human stem cell models of α-syn overexpression exhibit deficiencies and show toxicity [244,245,246]. Constitutive overexpression of α-syn in human progenitors from the human fetal cortex show an impaired switch to gliogenesis at later passages [247], and neuroprogenitors from an SNCA triplication case exhibited deficiencies in cellular energy metabolism and stress resistance [248]. α-syn overexpression in human iPSC neurons from a patient with a *SNCA* genomic triplication showed delayed neuronal differentiation and exhibited an altered neuronal network activity measured by patch clamping [249].

Remarkably, a lack of endogenous α-syn also causes deficits in adult neurogenesis. In α-syn KO mice [165], doublecortin (Dcx)-positive neuroblasts migrating from the SVZ along the rostral migratory stream (RMS) and the core of the OB were found to be reduced [237]. Calbindin- and calretinin-positive periglomerular neurons and TH-positive periglomerular OB neurons were decreased in α-Syn KO mice [237]. It has also been observed that this lack of α-syn causes a reduction in mushroom spines, indicating a lack of maturation [250]. A study assessing the stemness of adult rodent progenitors lacking α-syn showed a premature senescence of progenitors similar to dopamine fiber degeneration in chronic MPTP models or stereotactic injection of α-syn fibrils into the SN. This phenotype was rescued by viral delivery of human α-syn into the SN or treatment with L-3,4-dihydroxyphenylalanine L-DOPA, which stresses that α-syn is a regulator and critical factor to sustain the neurogenic potential of adult progenitors [251].

In summary, both α-syn loss and overexpression of wild-type and mutant α-syn impairs neurogenesis and it will be important to further understand the underlying genes and mechanisms involved in adult neurogenesis that lead to PD and ASD.

## 7. Key Concepts and Future Directions: Synaptic Dysregulation and Impaired Neurogenesis Are Common Links between LBD and ASD

While ASD as a neurodevelopmental disorder has long been linked to synaptic pathophysiology and neurogenesis, the root cause of neurodegenerative disorders has generally been attributed to the accumulation of proteins like α-syn and impairment of cellular functions, such as mitochondrial and lysosomal or proteasomal dysfunction [113]. With the identification of new causal genes that are involved in endosomal function and new emerging studies of known genes in the context of synaptic function, in particular α-syn, it is becoming evident that neurodevelopmental and neurodegenerative diseases share genes and converging pathways, such as an impairment of synaptic function resulting in an imbalance of neuronal circuits, and clinical motor and behavioral deficits [252,253,254,255,256]. It is also emerging that during the prodromal phase of PD/LBD, synaptic function is affected even before overt cell loss of susceptible neuronal populations occurs [254,257,258,259]. Synaptic deficits in LBD have been attributed to synaptic accumulation of α-syn at presynaptic terminals, affecting neurotransmitter release [260,261,262]. While synaptic function is one of the prominent findings that could link PD to ASD, there are also recent studies that highlight α-syn’s role in the nucleus, which may also connect these two disease entities [263].

Several aspects will need further attention in future studies of basic biology and clinical research to advance this field: First, clinical studies would benefit from including scales and testing of additional behavioral and motor symptoms in PD and ASD as these conditions show substantial overlap, which might have implications for clinical practice and future medical care. In addition, a multidisciplinary team comprising of neurologists and psychiatrists would better serve these patient populations. Second, genetic and genomic studies could expand screening in neurodegenerative disease, including high-resolution arrays or optical mapping for CNV analyses combined with mechanistic studies and pre-clinical models. Third, known genes, discussed in this review, with overlapping clinical symptoms should be further evaluated for disease-specific phenotypes and molecular changes in pre-clinical in vitro and in vivo disease models. Fourth, for both α-syn overexpression and deletion murine models, developmental studies and behavioral phenotypes will explain gain-of-function and loss-of-function deficits of α-syn and other genes.

In conclusion, we attempted to re-conceptualize clinical and experimental approaches for neurodegenerative and neurodevelopmental disorders to provide a new angle into disease genes and mechanisms that link neurodevelopmental disorders to neurodegeneration and even lay the foundation for the development of novel therapies.

## Figures and Tables

**Figure 1 ijms-21-05724-f001:**
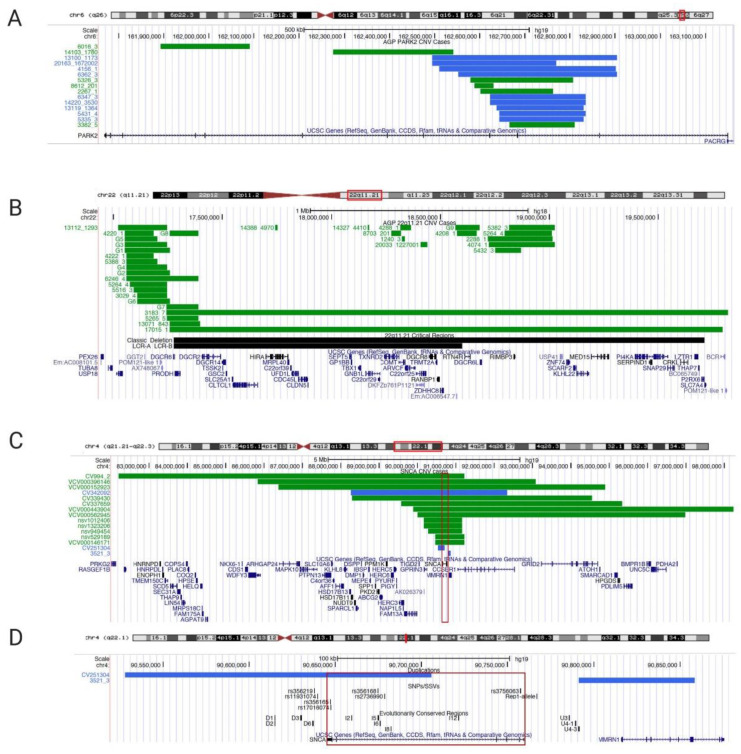
University of California Santa Cruz (UCSC) Genome browser view of copy number variants CNVs in *PARK2*, 22q11.2 deletion, and *SNCA/*4q22.1. (**A**) Partial exonic *PARK2* CNV deletion/duplication coordinates for UCSC genome browser custom tracks (chr6q26; genome build GRCh37/hg19, February 2009); green bars depict deletions, blue bars depict duplications; case identifiers listed on the left) on chromosome 6q26 found in the Autism Genome Project (AGP) cohort. Appendix A lists genomic positions to build custom tracks in UCSC Genome Browser. (**B**) 22q11.21 deletions’ coordinates for UCSC genome browser custom tracks (genome build NCBI36/hg18, March 2006). As a reference in black bars, we included the 22q11.21 deletion region (Velocardiofacial/DiGeorge syndrome, https://decipher.sanger.ac.uk/syndrome/16#genotype/cnv/21/browser) and the recently characterized critical region for a higher rate of autism (LCR-A to LCR-B) [75]. Case data were analyzed from AGP [73] for chromosome 22q.11.21. For cases with identical deletions, we grouped cases into groups G1 to G9 (column ID/Group ID). Every green bar depicts a deletion case with a numerical identifier; multiple cases are shown per line for space considerations. Appendix A lists genomic positions for custom tracks. (**C**) 4q22.1deletion cases with neurodevelopmental delay and/or ASD (<10MB; chr4:82,210,925-98,235,479); (**D**) Smaller *SNCA* duplication cases mapped to regulatory elements (SNPs/SSVs and evolutionary conserved regions) in the UCSC genome browser (chr.4:90,527,679-90,858,538; genome build GRCh37/hg19, February 2009). Every case has a numerical identifier, the red box indicates *SNCA* genomic region. Regulatory elements are listed in black in panel B (reviewed in [59]). Appendix A lists the genomic positions to build UCSC Genome Browser custom tracks.

**Figure 2 ijms-21-05724-f002:**
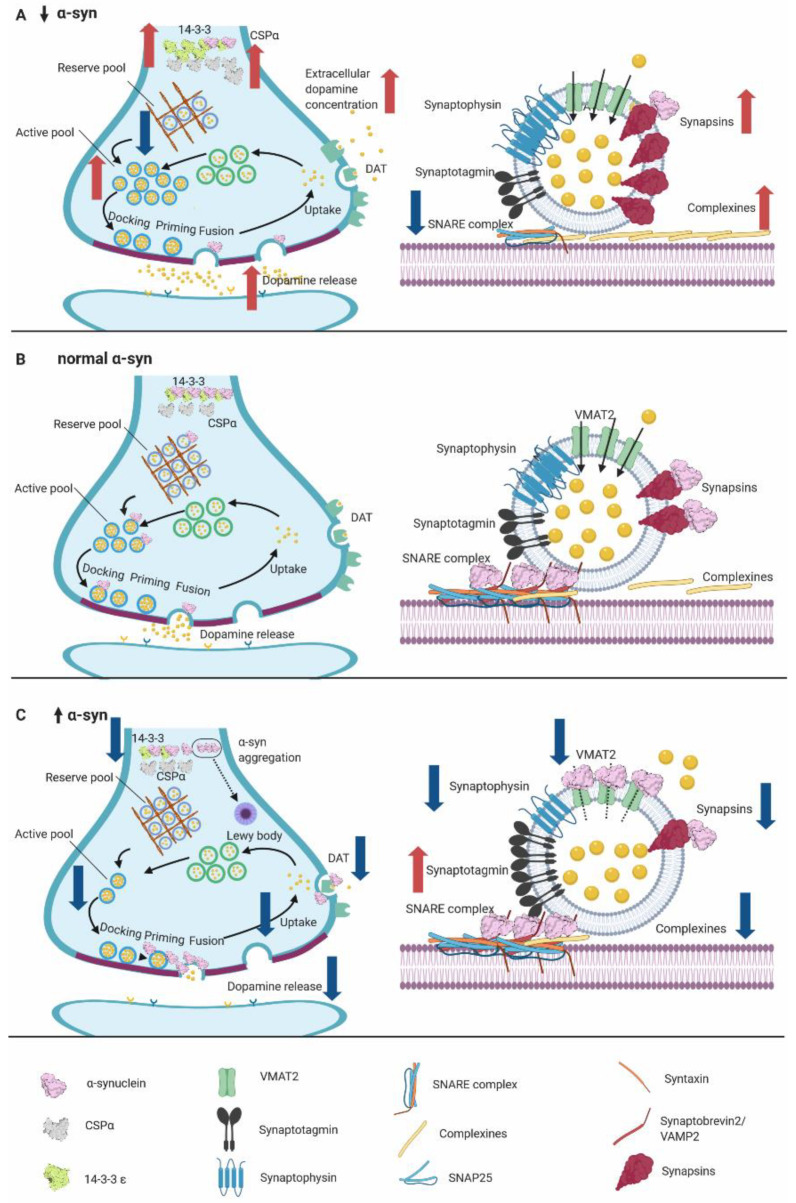
Overview of α-syn gene dosage effects at the synapse in the presynaptic terminal (upper panel) and synaptic vesicles (bottom). Red arrows represent an increase and blue arrows represent a decrease of protein or activity. (**A**) Alterations at the synaptic button due to α-syn downregulation/knockout (KO). Double and triple synuclein models showed an increase of 14-3-3ε, cysteine-string protein alpha (CSPα ) and complexines along with a decrease of 14-3-3ζ. Absence of α-syn can impair dopamine transporter (DAT) function, thus increasing extracellular dopamine levels and reduced availability of vesicles in the reserve pool, causing more vesicles to be released compensatorily and increasing dopamine release. Changes in the synaptic vesicle (bottom panel) indicate a decrease in Soluble N-ethylmaleimide sensitive factor attachment protein receptor proteins (SNARE) complex and Vesicle-associated membrane protein-2 (VAMP2) (member of the SNARE complex) along with an increase of synapsins (synapsin II, IIb, and III), and complexines (synaphin 1 and 2). (**B**) Normal levels of α-syn regulate physiological intracellular and extracellular levels of dopamine (upper panel), where it modulates the release cycle of synaptic vesicles, maintaining vesicle pools (active and reserve), and releasing an adequate concentration of dopamine to maintain the homeostasis of the synapses. In the bottom panel, α-syn interacts with the SNARE complex, promoting its assembly and interaction with complexines and synapsins. (**C**) Elevated or mutated α-syn leads to a reduction of dopamine release, along with dopamine uptake, which leads to a reduction of the active vesicle pool. An impairment of DAT alters the extracellular concentration of dopamine. In the synaptic vesicle (bottom panel), elevated or mutated α-syn generates a decrease in VMAT2 activity and mediates a decrease of synaptic proteins like synapsins (synapsin II, IIb), synaptophysin, and complexines. It also causes an elevated concentration of synaptotagamin in synaptic vesicles.

**Figure 3 ijms-21-05724-f003:**
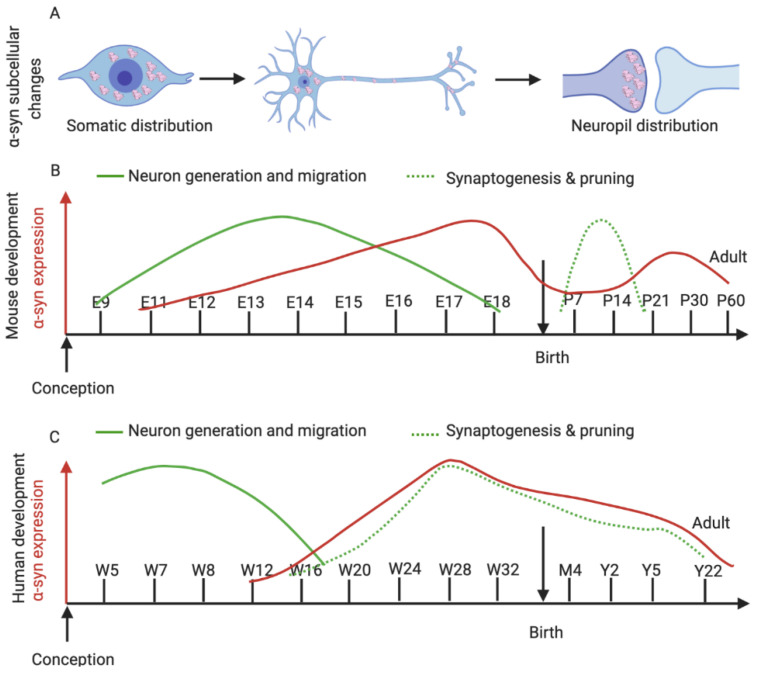
α-syn expression during neurodevelopment in mouse and human. (**A**) Localization of α-syn from a somatic distribution at early stages of development to neuropil expression late in development and during adulthood. (**B**) Timeline and schematic representation of the overall α-syn expression in the prenatal and postnatal (striatum) mouse brain. [203,205,206] (**C**) Timeline and schematic representation of the [205] overall α-synuclein expression in the developing human brain [210,211,214]. Red line represents α-syn expression; green line represents neuronal development and migration; green dotted line represents synaptogenesis and pruning. Mouse embryonic days (E); mouse postnatal day (P); gestational weeks (W), postnatal months (M), and postnatal years (Y).

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
