# Peer review of "The Role of Alpha-Synuclein and Other Parkinson’s Genes in Neurodevelopmental and Neurodegenerative Disorders"

_ijms, 2020, doi:10.3390/ijms21165724_

Round 1

Reviewer 1 Report

In clinical practice, the two diseases, autism spectrum disorder (ASD) and Parkinson's disease (PD), have different clinical treatments and are also divided into different disease categories (Psychiatry and Neurology). In my opinion, in addition to the analysis of genes, we should also analyze the common characteristics of the two diseases from the perspective of molecular pathology, such as the inflammatory cascade and mitochondrial dysfunction.This review does provide us with a lot of information about these two diseases from the perspective of genetic analysis. In clinical practice, the commonality of these two diseases can be understood at the early genetic level, and the impact and influence of future medical care should be discussed, especially when these two types of patients are treated by psychiatric team and neurology team. Since these two diseases seem to share certain characteristics, how likely is it to prevent the progression of individual diseases and even evolve each other? Is it possible for molecular therapies targeting gene and cell therapy to inhibit or attenuate individual diseases or evolve each other?

Author Response

We thank reviewer 1 for the thoughtful review and valuable suggestions on the clinical aspects of overlapping molecular features and translation into clinical practice.

We expanded two sections of the review to point out the importance of interdisciplinary clinical care between neurology and psychiatry. We also point out that there might be other molecular aspects that could be shared between autism and PD including inflammation and mitochondrial dysfunction and highlighted that there could be shared therapeutic approaches.

We modified and added to the following sections: lines 133-141:

“In summary, we describe striking overlapping clinical symptoms of parkinsonian motor problems in ASD and psychiatric symptoms such as depression, anxiety, and impulsive and compulsive activity in ASD and PD. Besides genetic causes, it will be important to assess overlapping molecular pathology of ASD and PD including inflammatory response and mitochondrial dysfunction that might even serve as early molecular biomarkers. In clinical practice, the commonality of these two diseases points towards the importance of an interdisciplinary clinical care team comprising of movement disorder specialists and psychiatrists for the clinical care and treatment of ASD and PD. This would allow not only serving patients better, but also gaining insight into common disease etiology and advancing the development of novel molecular treatments.”

We also modified the concluding paragraph lines 638-653, changes are indicated as underlined text:

“Several aspects will need further attention in future studies of basic biology and clinical research to advance this field: First, clinical studies would benefit from including scales and testing of additional behavioral and motor symptoms in PD and ASD as these conditions show substantial overlap, which might have implications for clinical practice and future medical care. In addition, a multidisciplinary team comprising of neurologists and psychiatrists would better serve these patient populations. Second, genetic and genomic studies could expand screening in neurodegenerative disease including high-resolution arrays or optical mapping for CNV analyses combined with mechanistic studies and pre-clinical models. Third, known genes, discussed in this review, with overlapping clinical symptoms should be further evaluated for disease-specific phenotypes and molecular changes in pre-clinical in vitro and in vivo disease models. Fourth, for both α-syn overexpression and deletion murine models, developmental studies and behavioral phenotypes will explain deficits of gain-of-function and loss-of-function of α-syn and other genes.

In conclusion, we attempted to re-conceptualize clinical and experimental approaches for neurodegenerative and neurodevelopmental disorders to provide a new angle into disease genes and mechanisms that link neurodevelopmental disorders to neurodegeneration and even lay the foundation for the development of novel therapies.”

Reviewer 2 Report

    Torres and colleagues summarize the role of several genes and genomic regions, including SNCA, PARK2, 22q11 deletion and FMR1 repeat expansion, in synaptic function and neurogenesis, and attempt to review the similarities and converging features in development of ASD and PD. This review is well-organized with detailed and sufficient materials, and offers a valuable view about the relationship of neurodevelopmental and neurodegenerative disorders. I have only few points to address:

  1. Line 47, page 2: ‘… increase which age;’ should be ‘… increase with age;’
  2. Line 142, page 5: ‘Table S1’ should be ‘Figure S1’.

Author Response

Thanks so much for the favorable review! We corrected the errors/typos and made the changes in the manuscript.

Of note, we actually meant Table S1 with the list of SNCA multiplication cases instead of Figure S1 which depicts methods for CNV detection.

Round 2

Reviewer 1 Report

All the points raised have been addressed.

Reviewer 2 Report

This is an exellent review, I approve of its publication.